# Research on the Efficiency of Working Status Based on Wearable Devices in Different Light Environments

**DOI:** 10.3390/mi13091410

**Published:** 2022-08-27

**Authors:** Shuhan Yan, Yuncui Zhang, Sen Qiu, Long Liu

**Affiliations:** 1The Research Institute of Photonics, Dalian Polytechnic University, Dalian 116034, China; 2Key Laboratory of Intelligent Control and Optimization for Industrial Equipment of Ministry of Education, Dalian University of Technology, Dalian 116024, China

**Keywords:** inertial sensor, posture reconstruction, working status measurement, gradient descent method, work efficiency, light environment

## Abstract

According to the working scenes, a proper light environment can enable people to maintain greater attention and meditation. A posture detection system in different working scenes is proposed in this paper, and different lighting conditions are provided for changes in body posture. This aims to stimulate the nervous system and improve work efficiency. A brainwave acquisition system was used to capture the participants’ optimal attention and meditation. The posture data are collected by ten miniature inertial measurement units (IMUs). The gradient descent method is used for information fusion and updating the participant’s attitude after sensor calibration. Compared with the optical capture system, the reliability of the system is verified, and the correlation coefficient of both joint angles is as high as 0.9983. A human rigid body model is designed for reconstructing the human posture. Five classical machine learning algorithms, including logistic regression, support vector machine (SVM), decision tree, random forest, and k-nearest neighbor (KNN), are used as classification algorithms to recognize different postures based on joint angles series. The results show that SVM and random forest achieve satisfactory classification effects. The effectiveness of the proposed method is demonstrated in the designed systematic experiment.

## 1. Introduction

With the development of scientific research on light health, it has been proved that the light environment affects not only eye health [1] but also work efficiency, emotion, and physiological activities [2]. Therefore, providing an intelligent light environment that adapts to different working scenes is emerging as a research hotspot. Some studies have shown that office workers would like to control the lighting autonomously to keep them motivated and focused throughout the day. Participants are more satisfied when they perceive higher lighting quality in their offices Meanwhile, a higher color temperature (7500 k compared to 3000 k) is proved to increase mental activity at 1000 lux [3]. The lower color temperature can reduce central nervous system activity [4].

Attention is the primary evaluation indicator to detect the effect of the light environment on work efficiency. Several methods for monitoring attention have been employed in previous studies. Wang et al. assessed subjects’ performance of attention influenced by illumination using the Schulte grid method [5]. However, the experimental process can be disrupted by this invasive and mandatory detection method. Roberto et al. designed a multimodal acquisition framework to monitor blink activity during online task execution for concentration assessment [6]. Landmark detection and ROI extraction were used to localize eye regions and blink detection. However, in an environment without visual feedback, this flexibility is lost, which makes it difficult to monitor the user’s attention. Lee et al. verified the effect of illuminance and associated color temperature of LED lighting on memory during work [7]. The 3-back task lasts 5 min, and the number of correct answers in the 3-back task reflects the attention level from the side. However, the data obtained by this method are very subjective and do not directly reflect the changes brought about by the light environment on human attention.

Nowadays, it is a standard method to monitor attention using Mindwave. This small instrument causes no discomfort to participants and shows real-time changes in the brain. Yan et al. demonstrated that the rate of the α-wave and β-wave of the students’ brain waves significantly differed at different color temperatures (2700 K, 4000 K, 6500 K) and illumination levels (300 lx, 750 lx, 1000 lx) [8]. As the color temperature and illumination increase, students became more excited and sensitive, showing a roughly positive correlation.

The different working states of people can be judged by observing their eye movements, expressions, postures, etc. We find that numerous impressive research results are obtained based on video methods of capturing human posture. Sanchez et al. investigated a video-based method to generate a numerical measure of postural alignment of the head and trunk in sitting [9]. The results show that the method is more accurate and reliable in obtaining a more detailed spine profile than subjective judgment. However, some tricky problems are often encountered in practice, such as user privacy issues, obstacle occlusion, and the influence of light, which reduce the accuracy of motion recognition. In addition, it is feasible to obtain the contour and motion state of the human spine or extremities using pressure pads. Pressure sensors are mounted on the seat surface or backrest to capture the participant’s pressure distribution for the identification of sitting posture. Shunsuke et al. measured the time series of the pressure distribution of a person on a pressure cushion to analyze the sitting person’s move direction and the amount [10]. Jawad Ahmad et al. designed a flexible and inexpensive screen-printed large-area pressure sensing system that eliminates erroneous pressure information and achieves more than 80% accuracy in sitting classification [11]. Nevertheless, the pressure distribution reflects only the rough sitting posture and cannot reconstruct the body posture. Some mechanical, optical and electromagnetic human posture capture systems have also been studied. Mechanical human posture capture systems limit human motion [12]. Optical-based human pose capture systems are more expensive, require a high-light environment, and are susceptible to occlusion [13]. The electromagnetic-based human posture capture system is susceptible to magnetic field interference, and magnetic field distortion caused by surrounding metal objects severely affects accuracy [14].

With the development of microelectromechanical systems (MEMS), sensors have become more accurate and compact, and human posture capture based on IMU is gradually becoming a research hotspot [15]. Shull et al. reviewed and evaluated the clinical applications of wearable gait sensing and feedback [16]. The inertial measurement unit is the most widely used wearable sensor, and its clinical benefits have been proven. Qiu et al. designed a wearable smart system without root node constraints for high-precision human motion capture [17]. Rubaiyat et al. investigated the four main daily activities of walking, sitting, walking up and down stairs using a real-time human activity recognition system based on a single triaxial accelerometer, with a recognition rate of more than 90% [18]. However, sensors worn only at specific locations fail to provide a comprehensive analysis of posture, resulting in missing parameters that affect the accuracy of experiments. The most commonly used sensor combinations in the early days were gyroscopes and accelerometers. Mahmoud El-Gohar used two wearable inertial measurement units to estimate the angle of the human shoulder and elbow. The orientation was calculated by integrating the angular velocity of the gyroscope. Accelerometers were used as compensation data to compensate for drift errors. Non-linear inertial posture fusion was achieved by applying the traceless Kalman filter [19]. The rapid drift of estimates after a few seconds was caused by time-varying biases and noise affecting accelerometer and gyroscope measurements, which are unreliable. Therefore, researchers investigated multi-sensor fusion to solve the drift problem. Bachmann et al. proposed a quaternion complementary filter based on accelerometers and magnetometers for compensating for the drift of gyroscope signals [20]. Yun et al. proposed a design combining extended Kalman filtering and Gauss–Newton methods for parameter optimization [21]. The quaternion is solved from the accelerometer and magnetometer measurements by the Gauss–Newton iterative method, and the extended Kalman filter fuses the quaternion and the angular velocity to obtain the posture.

Currently, traditional methods for monitoring and evaluating posture rely on many laboratory-based motion capture systems that lack application in realistic scenarios outside the laboratory. Future research should focus on the natural human environment with continuous and long-term monitoring and interventions for motor posture. Previous research has been applied in the medical, sports and virtual reality fields, but few researchers have applied wearable inertial sensors in working conditions. In this paper, long-term human posture monitoring is performed in a natural working environment by a self-made posture detection system based on micro inertial sensors. Our purpose is to analyze participants’ attention in different states and light environments, and this paper mainly focuses on the following points.

An association between body posture in different working scenes and optimal light conditions is established to enhance the work efficiency.A wearable device is designed with multiple IMUs fixed on the body to monitor human posture in quaternion format. The full description of hardware and algorithms is presented. Experimental results reveal that the scheme can effectively recognize human postures in various work tasks.Assessment of brain activity under different light conditions at various degrees of attention and meditation shows that environmental lighting factors are crucial for improving work efficiency.

## 2. Materials and Methods

### 2.1. System Platform and Data Collection

This system aims to investigate a methodology for improving the work state based on wearable devices. Musculoskeletal disorders can be triggered in office occupations due to prolonged sedentary behavior [22]. In this paper, four common work postures are considered. They correspond to four specific tasks: reciting English words, performing English reading comprehension, browsing essays, and listening to music. As shown in Figure 1, the system platform architecture shows that a whole-body posture capture system based on inertial measurement units (IMUs) is used to capture and reconstruct the human posture. The human posture data are learned and trained locally by machine learning classification algorithms, and four types of working scenarios (memory, thinking, understanding, and leisure) are classified. The features of the four work tasks are uploaded to the cloud platform, where the office lighting system is unified and regulated. Providing the best light environment according to the working posture in different work scenes will improve people’s attention and meditation, obtaining the best office state.

We develop an IMU-based whole-body posture capture system to collect raw sensor data of participants’ working postures and reconstruct human activities. The system hardware and control platform UI are shown in Figure 2. This system consists of several miniature inertial sensor nodes (IMU), a transceiver and computer software. Each inertial sensor node consists of a ICM-20948 nine-axis inertial sensing module (a three-axis accelerometer, a three-axis gyroscope and a three-axis magnetometer), a Lora communication module and a trans-flash (TF) memory card. User comfort and impact on motion are our special concerns, and we try our best to reduce the size of the inertial sensor node. The final size of the PCB is 22 × 24 (mm), and the 3D housing size is 31.5 × 29.1 × 25.9 (mm). Although the nodes are worn on the limbs via nylon straps, participants experience no discomfort and move freely, and no harmful effects of the devices are observed on the human body. ICM-20948 is encapsulated in a 3 × 3 × 1 mm QFN package, significantly reducing chip size and power consumption while improving performance and reducing costs. Table 1 shows the sensor specifications. The battery endurance of the self-made inertial sensor node is about 3 h when fully charged, which is sufficient to satisfy the experimental requirements. Based on Lora wireless communication, the node control program sends working commands to the sensor nodes through the transceiver and monitors the working status of the nodes. The sensor nodes immediately acquire the participant’s posture data with a sampling frequency of 400 HZ (up to 800 Hz) when the software sends the “Start Acquisition” signal. All collected data are stored in the TF cards with a flash file system in nodes. The data in TF cards are converted into CSV file by the node control program, and then the whole-body motion postures of the participant are reconstructed by the Matlab motion analysis program. The framework of the posture capture system is shown in Figure 3.

In order to analyze the effect of different light environments on attention and meditation under different work tasks, the MindWave EEG monitoring system is used to monitor the attention and meditation of the human brain under different light environments [23]. The EEG monitoring system consists of a MindWave brainwave device, a computer or smartphone software. The brain produces electrical signals, which we commonly call brainwaves and which are captured by MindWave through ThinkGear technology with a sampling frequency of 512 Hz. The 50 Hz–60 Hz AC noise filter can filter, amplify, A/D convert and output the collected signals. MindWave enables non-invasive dry electrode technology. Unlike traditional EEG signal collectors, it is simple to operate, consumes less power, is lightweight and easy to wear. The specifications of MindWave are shown in Table 2. The shape of MindWave is shown in Figure 4. The EEG signals are measured by a reference electrode contact placed in the earlobe and an electrode device placed in the forearm of the forehead. These signals are processed by an integrated chip. The raw EEG data can be output as delta, theta, low beta, high beta, low alpha, high alpha, low gamma and high gamma waves. eSense is a patented algorithm based on ThinkGear technology. The quantified eSense index values, i.e., “Attention” and “Meditation”, are obtained to reflect people’s mental state. Bluetooth is the medium of communication between MindWave and a computer or a mobile phone. The collected data is processed and saved in a specified data file.

### 2.2. Definition

#### 2.2.1. Structural Model of the Human Body and Joint Angles

Based on human anatomy, the human structural model is defined as a rigid structure consisting of 15 limb segments (head, spine, waist, left upper arm, left forearm, right upper arm, right forearm, left shoulder, right shoulder, left thigh, left calf, right thigh, and right calf) [24], as shown in Figure 5a. Figure 5b shows that fifteen nine-degree-of-freedom inertial sensor nodes are placed on the corresponding limb segments, and each node represents the corresponding rigid limb segment. The motion postures of the head, left shoulder, right shoulder, and chest are captured by one node placed on the chest. Nodes on the limb segments are used to obtain raw acceleration, gyroscope, and magnetometer information during data acquisition.

Each two adjacent limb segments constitute a joint, and the joint angle corresponds to the flexion of the joint. Sitting on a desk, the upper limbs are more involved in postural adjustments. The upper extremity has a greater range of motion. The flexion angles of the upper and lower limbs are defined as the breast flexion angle (BF), waist flexion angle (WF), shoulder flexion angle (SF), elbow flexion angle (EF), and knee flexion angle (KF) [25], as shown in Figure 6.

#### 2.2.2. Definition of Coordinate Systems

The raw data of the motor posture are obtained from the IMUs on the extremities. The posture calibration and conversion need to be performed in three coordinate systems to reconstruct the participants’ motion posture based on the IMU data, as shown in Figure 7. The three coordinate systems are defined as follows.

Geodetic coordinate system (GCS): This system is also known as the navigation coordinate system. The three axes are perpendicular to each other, pointing north (x-axis), east (y-axis) and to the ground (z-axis), corresponding to the ENU coordinate system.Sensor coordinate system (SCS): It is defined as the coordinates of the sensor nodes placed on the human body.Body coordinates (BCS): As shown in the figure, we take the pelvis as the origin, with the X-axis perpendicular to the body surface outward, the Y-axis perpendicular to the X-axis to the right, and the Z-axis downward.

### 2.3. Methodology

This section describes the structure of the motion analysis program; the framework of the motion analysis program is shown in Figure 8. The magnetometer eliminates errors by ellipsoidal fitting. Accelerometers and gyroscopes are used for signal pre-processing. Initial posture quaternions are obtained from the magnetometer and accelerometer and used for initial state estimation. The gradient descent method is used to fuse multi-sensor data. The initial posture quaternion is compensated by the gyroscope, and finally an approximate exact solution of the posture quaternion is obtained for motion posture reconstruction and joint angle calculation.

#### 2.3.1. Sensor Calibration

In the gyroscope error model, $\omega_{m(t)}$ and $\omega_{s(t)}$ represent the measured and actual values of angular velocity, respectively, at time *t*, $\beta_{\omega ,s}$ represents the gyroscope misalignment, and $\mu_{\omega ,s}$ represents the white noise interference of the gyroscope. In the accelerometer error model, $a_{m(t)}$ and $a_{s(t)}$ represent the measured and actual values of acceleration, respectively, at time *t*, $g_{s}$ is the gravitational acceleration component, and $\beta_{a,s}$ represents the white noise interference of the accelerometer.

The calibration of the magnetometer is required because the soft and hard iron of the surrounding environment distorts it. In this paper, the ellipsoid fitting method is used to calibrate the magnetic field [26]. The ellipsoidal fitting equation is shown below:(1)$$\frac{{\left(H_{m}^{x}-e_{h}^{x}\right)}^{2}}{a^{2}}+\frac{{\left(H_{m}^{y}-e_{h}^{y}\right)}^{2}}{b^{2}}+\frac{{\left(H_{m}^{z}-e_{h}^{z}\right)}^{2}}{c^{2}}=R^{2}$$
where $e_{h}^{x}$, $e_{h}^{y}$, $e_{h}^{z}$ are the deviations caused by hard iron errors, and $H_{m}^{x}$, $H_{m}^{y}$, $H_{m}^{z}$ are the data measured by the three-dimensional magnetometer. *a*, *b*, and *c* are the lengths of the ellipsoidal semiaxes, and *R* is the modulus constant of the earth’s magnetic field. In this paper, the ellipsoidal fitting method is used to eliminate ferromagnetic interference. The calibration outcome of the magnetometer is shown in Figure 9.

#### 2.3.2. Initial State Estimation

In this paper, quaternions are used as state variables to represent the 3D rotation information of each limb segment to avoid the general locking problem of Euler angles, and quaternions are lighter than matrices. The quaternion is defined as follows.
(2)$$q=\left(q_{0},q_{1}i,q_{2}j,q_{3}k\right)$$
where $q_{0}$, $q_{1}$, $q_{2}$, and $q_{3}$ are real numbers, and *i*, *j*, and *k* are three-dimensional space unit vectors. This quaternion represents the working posture of the participants in the GCS. Since the position of the sensor on the corresponding limb changes at any time during the actual measurement, the initial state estimation ensures that the transition relationship between the sensor and the corresponding limb component is a fixed value. Participants stand with their arms naturally down, facing north for a few seconds. In BCS, the X-axis points north, the Y-axis points east, and the Z-axis is perpendicular to the ground so that the BCS in the initial posture overlaps with the GCS. Using the following equations, the Euler angles are calculated from the magnetometer and accelerometer.
(3)$$\begin{array}{c} \phi_{in}=\arctan\left(a_{S}^{y},a_{S}^{z}\right)\\ \theta_{in}=\arcsin\left(-a_{S}^{x}/g\right)\\ h_{G}^{x}=h_{S}^{x}\cos\theta_{in}+h_{S}^{y}\sin\theta_{in}\sin\phi_{in}+h_{S}^{z}\sin\theta_{in}\sin\phi_{in}\\ h_{G}^{y}=h_{S}^{y}\cos\phi_{in}-h_{S}^{z}\sin\phi_{in}\\ \varphi_{in}=-\arctan\left(h_{G}^{y}/h_{G}^{x}\right) \end{array}$$
where $\phi_{in}$, $\theta_{in}$, and $\varphi_{in}$ represent the angle of roll, pitch and yaw, respectively, *g* represents the gravitational acceleration, $a_{S}^{x}$, $a_{S}^{y}$ and $a_{S}^{z}$ represent the accelerometer measurements, and $h_{S}^{x}$, $h_{S}^{y}$ and $h_{S}^{z}$ represent calibrated magnetometer measurements, respectively. The initial quaternion conversion matrix $q_{Sin}^{G}$ from SCS to GCS is obtained using the interconversion between Euler angles and quaternions, as shown in Equation (Equation 4).
(4)$$q_{Sin}^{G}=\left[\begin{array}{c} \cos\left(\frac{\phi_{in}}{2}\right)\cos\left(\frac{\theta_{in}}{2}\right)\cos\left(\frac{\varphi_{in}}{2}\right)+\sin\left(\frac{\phi_{in}}{2}\right)\sin\left(\frac{\theta_{in}}{2}\right)\sin\left(\frac{\varphi_{in}}{2}\right)\\ \sin\left(\frac{\phi_{in}}{2}\right)\cos\left(\frac{\theta_{in}}{2}\right)\cos\left(\frac{\varphi_{in}}{2}\right)-\cos\left(\frac{\phi_{in}}{2}\right)\sin\left(\frac{\theta_{in}}{2}\right)\sin\left(\frac{\varphi_{in}}{2}\right)\\ \cos\left(\frac{\phi_{in}}{2}\right)\sin\left(\frac{\theta_{in}}{2}\right)\cos\left(\frac{\varphi_{in}}{2}\right)+\sin\left(\frac{\phi_{in}}{2}\right)\cos\left(\frac{\theta_{in}}{2}\right)\sin\left(\frac{\varphi_{in}}{2}\right)\\ \cos\left(\frac{\phi_{in}}{2}\right)\cos\left(\frac{\theta_{in}}{2}\right)\sin\left(\frac{\varphi_{in}}{2}\right)-\sin\left(\frac{\phi_{in}}{2}\right)\sin\left(\frac{\theta_{in}}{2}\right)\cos\left(\frac{\varphi_{in}}{2}\right) \end{array}\right]$$

The initial quaternary transformation from SCS to BCS is the same as the quaternary transformation from SCS to GCS, i.e., $q_{S_{in}}^{B}\approx q_{S_{in}}^{G}$. The quadratic transformation relation from BCS to GCS represents the working posture of a participant in GCS, which can be expressed as $q_{B_{in}}^{G}=q_{S_{in}}^{G}⊗q_{B}^{S}$ and $q_{B}^{S}={\left(q_{S}^{B}\right)}^{*}$, where * represents the conjugate matrix. Since the sensors are fixed on the surface of the limbs, the rotation relation from SCS to BCS is assumed to be constant, i.e., $q_{S}^{B}\approx q_{S_{in}}^{B}$. Thus, the quadratic rotational transformation of the limbs with respect to the ground is $q_{B_{in}}^{G}=q_{S_{in}}^{G}⊗{\left(q_{S}^{B}\right)}^{*}$.

#### 2.3.3. Posture Update Algorithm

Qiu et al. review the current state of research and challenges in sensor information fusion theory for human behavior recognition applications [17]. Among the information fusion algorithms, the extended Kalman filter [27], complementary filter [28], and particle filter [29] are commonly utilized. The gradient descent (GD) method is widely used [30]. Gyroscope measurements constitute the basis of posture reconstruction, while accelerometers and magnetometers are used to estimate gyroscope errors. The gyroscope, accelerometer, and magnetometer are complementarily fused to solve the posture quaternion with minimum error. The optimal value of the rotational transform quaternion $q_{S}^{G}$ is obtained by the gradient descent method, and the rotational error is eliminated while capturing the working postures.

The error function between the field direction $n^{G}$ of the sensors in the GCS and the direction $n^{S}$ in the SCS is defined as follows.
(5)$$f\left(q_{S}^{G},n^{G},n^{S}\right)={\left(q_{S}^{G}\right)}^{*}⊗n^{G}⊗q_{S}^{G}-n^{S}$$

In order to make $f\left(q_{S}^{G},n^{G},n^{S}\right)$ converge to zero and obtain the optimal solution for $q_{S}^{G}$, we use the gradient descent algorithm proposed by Madgwick [31] to solve this issue. Firstly, Equation (Equation 6) calculates the gradient of the error function:(6)$$\nabla f\left(q_{S}^{G},n^{G},n^{S}\right)=J^{T}\left(q_{S}^{G},n^{G}\right)f\left(q_{S}^{G},n^{G},n^{S}\right)$$
(7)$$q_{S}^{G}\left(t\right)=q_{S}^{G}\left(t-1\right)-\xi \frac{\nabla f\left(q_{S}^{G},n^{G},n^{S}\right)}{∥\nabla f\left(q_{S}^{G},n^{G},n^{S}\right)∥}$$

The sensor’s rotational quaternion $q_{S}^{G}$ at the current moment is obtained through several iterations by calculating Equation (Equation 7) until the final moment. This value represents the local extremum, which has changed since the initial state. The error of the rotation matrix is minimized by using the gradient descent method.

#### 2.3.4. Movement Posture Reconstruction and Joint Angle Calculation

As mentioned earlier, the human body is defined as a rigid structure consisting of limb segments. People change their work posture to adapt to various work scenarios. In motion posture capture, the pelvis is used as the iterative origin of posture construction, and the iterative procedure can obtain all limb postures.

We take the skeleton of the right leg as an example. We define thigh and calve segments as *V* and *W*, respectively, as shown in Figure 10. The limb segment *V* is the parent segment, the limb segment *W* is the child segment, and the posture of *W* is derived from *V*. $D_{V_{0}}$ and $D_{V_{1}}$ are the starting and ending positions of the thigh limb segment *V* in the GCS. $D_{W_{0}}$ and $D_{W_{1}}$ are the starting and ending positions of the calf limb segment *W* in the GCS. At time *t*, $D_{V_{1}}\left(t\right)$ equals $D_{W_{0}}\left(t\right)$. $l_{v}$ and $l_{w}$ are defined as vectors of limb lengths.

As mentioned above, the quadratic rotation of each limb segment from BCS to GCS can be expressed as:(8)$$q_{B}^{G}=q_{S}^{G}⊗q_{B}^{S}$$

Therefore, the position $D_{V_{1}}\left(t\right)$ of the end of the *V* segment and the position $D_{W_{1}}\left(t\right)$ at the beginning of the *W* segment in the GCS are:(9)$$D_{V_{1}}\left(t\right)=D_{V_{0}}\left(t\right)+q_{B,V}^{G}\left(t\right)⊗\left[0\;l_{V}^{G}\left(t\right)\right]⊗{\left(q_{B,V}^{G}\left(t\right)\right)}^{*}$$
(10)$$D_{W_{1}}\left(t\right)=D_{W_{0}}\left(t\right)+q_{B,W}^{G}\left(t\right)⊗\left[0\;l_{W}^{G}\left(t\right)\right]⊗{\left(q_{B,W}^{G}\left(t\right)\right)}^{*}$$
where $q_{B,V}^{G}\left(t\right)$ and $q_{B,W}^{G}\left(t\right)$ are the rotational quaternions of limb segments *V* and *W* from BCS to GCS at time *t*. Thus, the whole-body motion postures are calculated by each limb through Equations (Equation 9) and (Equation 10) with multiple iterations from the origin.

The joint angles are formed by the joints of adjacent limb segments [32]. As shown in Figure 10, the joint angle $\theta_{KF}$ formed by limb segments *V* and *W* can be calculated by the following equation.
(11)$$\theta_{KF}=\arccos\frac{\vec{d_{V}^{G}\left(t\right)}\cdot \vec{d_{W}^{G}\left(t\right)}}{\left|\vec{d_{V}^{G}\left(t\right)}\right|\left|\vec{d_{W}^{G}\left(t\right)}\right|}$$

As a result, the joint angle of the whole body posture can be solved by the inverse cosine between the vectors of two adjacent limb segments.

## 3. Experiment

### 3.1. Experimental Environment

The basic lighting experiments were performed in the Optical Experiment Base of Dalian Polytechnic University. The laboratory was set up as an office with no natural light. The experimental environment is shown in Figure 11. Monitors were placed directly in front of participants, with a distance of 50 cm between the monitor and the eyes. The chair height was set according to the participants’ height to satisfy the participants maintaining a knee angle of 90 degrees. In order to study the effect of different lighting environments on different working conditions, the experiment provided four color temperature lighting combinations: (1) 6000 K–600 lx, (2) 6000 K–500 lx, (3) 5000 K–500 lx, and (4) 3000 K–300 lx.

### 3.2. Experimental Process

To avoid abnormal sitting due to impaired vision, we recruited 6 participants (two female and four male) with a visual acuity score greater than 0.5 (wearing glasses) who are in good physical condition and volunteered to participate in the study. Before the experiment, participants received enough sleep and performed 2 h of relaxation. Consumption of alcoholic beverages, coffee and other stimulants, and consciousness-altering substances was prohibited. In addition, none of the subjects suffered from neck, shoulder, leg or arm pain.

Participants wore self-made garments and ten nodes on the relevant limb segments (breast, pelvis, upper and lower arms, thighs, and calves) during the experiment, as illustrated in Figure 12. Sensor nodes recorded the whole body’s posture movement. The MindWave brain-computer interface device was used to record brain wave data. Participants were asked to sit at a computer table and complete an experimental task prompted by the computer screen within a specific time.

Participants wearing the sensor nodes completed the following four scenarios sequentially according to their habits, each lasting 5 min. The participants carried out the following tasks, as shown in Figure 12: (1) recite English words, (2) do English reading comprehension, (3) browse essays, and (4) listen to music. We assume that reciting English words represents a memory-type work scenario requiring a high workload and great attention. Doing test papers represents a thinking-type work scenario requiring some attention. Browsing essays requires concentration but not much workload and represents a comprehension-type work scenario. Finally, listening to music is different from the other states. It represents a kind of relaxation in work, which can help participants relax and free themselves from other work scenes. Each task that maintains 20 min was randomly ordered. Participants perform the task in four lighting combinations. The obtained posture data were used for machine learning to identify different work scenarios.

The experiment was divided into 4 × 4 experimental conditions, as shown in Table 3. The participants’ EEG data corresponding to different experimental conditions were also recorded at the same time as the posture acquisition.

## 4. Algorithm Verification and Experimental Results

### 4.1. Verification of the Whole Body Pose Capture Algorithm

To compare the accuracy of a self-made whole-body posture capture system with an optical capture system in the laboratory, we used an optical capture system produced by Natural Point, USA. This optical capture system uses near-infrared imaging to analyze and reconstruct the three-dimensional motion of the human body with a displacement accuracy of up to 0.1 mm. In the comparison test, 12 infrared cameras were used. Participants were required to wear specific clothing. Optical markers and inertial sensor nodes were placed at specific locations on their bodies. Participants walked around the laboratory at will while two systems simultaneously performed posture capture. The experimental scenario is shown in Figure 13. The optical capture system and the inertial posture capture system were used to collect the posture data of the subjects at the same time with sampling frequencies of 120 Hz and 400 Hz. The third-order Hermite interpolation method was used to supplement the data collected by the optical capture system to align the data of the two systems.

The joint angles $\theta_{EFl}$ and $\theta_{EFr}$ of the upper limbs were used as examples to compare the results of joint angle. The comparison curve between the optical capture system and the IMU-based inertial posture capture system is shown in Figure 14. The error distribution of the joint angle of the two systems was described as the normal distribution fitting curve and frequency histogram, as shown in Figure 15. The correlation coefficients of our method and the optical system are 0.9983 and 0.9813, respectively. The results show that our method can produce the same results compared to the optical system, which verifies the reliability of the performance of the self-made whole-body pose capture system.

### 4.2. Posture Reconstruction and Work Status Classification

In daily work, the combination of upper and lower limb movements constitute our different sitting postures. In order to provide a suitable lighting environment for different postures, the postures under different lighting conditions were reconstructed by the motion analysis program, as shown in Figure 16. Using machine learning classification algorithms, the participant’s work tasks were identified based on the joint angle features of the reconstructed postures.

#### 4.2.1. Dataset Segmentation

The participants’ joint angles ($\theta_{\mathrm{BF}}$, $\theta_{\mathrm{WF}}$, $\theta_{\mathrm{SF}}$, $\theta_{\mathrm{EF}}$, and $\theta_{\mathrm{KF}}$) were calculated by the above method. Every 1000 ms, the time series of each joint angle are separated into multiple time window sequences. In order to expand the data sample size, 50% of data overlaps were set between adjacent windows. As an example, word recitation data for each joint angle feature were segmented into 178 groups. These 178 lines of data for reciting words were labeled as 1, and so on; the data for doing English reading comprehension were labeled as 2, the data for browsing essays were labeled as 3, and the data for listening to music were labeled as 4.

#### 4.2.2. Feature Extraction

The segmented data cannot be used directly in the classifier’s classification. It is essential to perform feature extraction on the data and extract the effective information reflecting the movement’s properties to produce a feature sequence, which will be used as the classifier’s input. We extracted 23 features from all joint angle sequences, including time-domain and frequency-domain data. Table 4 lists the descriptions of the 23 features.

#### 4.2.3. Performance on Different Classification Methods

Five classical machine learning algorithms, including logistic regression, support-vector machine (SVM), decision tree, random forest, and K-nearest neighbor (KNN), were used as classification methods to identify different postures based on sequences of joint angles. K-fold cross-validation [33] was used to partition the dataset. Each subset was disjoint and equal in size. One of the K copies was selected as the validation set, and the remaining K-1 copies were used as the training set. In this paper, the 10-fold crossover was applied to divide the dataset, and the remaining data were combined into several different training and validation sets, excluding the test set. The feature data in the test set never appeared in the training and validation sets. The feature data of the postures that have appeared in the training set may become samples in the validation set next time. This ensures that all participants’ feature data are involved in training, validation and testing. Finally, the average of each validation error was used as the final validation error of the model. The model that obtained the highest accuracy on the validation dataset was selected for evaluation on the test set.

We evaluate the classifiers by four classification model evaluation metrics: accuracy, precision, recall, and F1-score. All the evaluation metrics in this paper are calculated using the scikit-learn machine package in python. Precision refers to the proportion of correctly classified results predicted as positive samples. Recall refers to the proportion of correctly classified results in the actual positive samples. The precision and the recall are related to each other. When the recall and precision of different models have their advantages, the F1-score can evaluate the model well, which is the harmonic average of precision and recall. As shown in Table 5, the recognition Accuracy of all models is above 95%, and other evaluation metrics such as precision, recall, and F1-score are also high. All the above evaluation metrics are weighted averages. KNN and random forest perform best in all algorithms. It also shows that joint angle is an effective method to describe human posture.

The final output of a classification model is often a probability value, and we generally need to convert the probability value to a specific category. When different models take different thresholds, the receiver operating characteristic (ROC) curve and precision recall curve (PRC) are effective metrics to evaluate the performance of classifiers. AUC is defined as the area under the ROC curve. The ROC curve does not explicitly state which classifier is better, but the classifier corresponding to a larger AUC is better. As shown in Figure 17a and the sixth row of Table 5, random forest, SVM and KNN perform better. Similar to the definition of AUC, AUPRC is the area under the PRC curve symbolizing the effectiveness of each classifier [34]. As shown in Figure 17 and the sixth and seventh row of Table 5, the random forest and SVM models always have better performance than other models. The evaluation results show that posture acquisition with our method can effectively classify different work tasks.

### 4.3. Analysis of Attention and Meditation in Different Light Environments

The “eSense” algorithm assesses a person’s current mental state through two indices of “Attention” and “Meditation”. The eSense index indicates the user’s level of attention and meditation with specific values between 1 and 100. A value between 40 and 60 represents a general range, which is similar to the “baseline” determined in conventional brainwave measurement techniques. A value between 60 and 80 represents a higher than average level of attention or meditation. A value between 80 and 100 represents a very high level of attention or meditation. Similarly, a value between 20 and 40 represents the “low-value area”, while a value between 1 and 20 represents the “lowest value area”.

“Brainwave Visualizer” is a software platform for displaying brainwaves developed by NeuroSky. This software views raw sensor data of brain waves, brain wave spectrum, “Attention”, and “Meditation”. Figure 18 shows the EEG signal, “Attention”, and “Meditation” of listening to music at 3000 k–300 lx.

The EEG data of participants under different experimental conditions are obtained using Mindwave. Figure 19 depicts the original attention data for the four tasks of (1) reciting English words, (2) performing English reading comprehension, (3) browsing essays, and (4) listening to music. From Figure 19a, the distribution of the attention curves for reciting English words under the four lighting settings can be seen, and the attention is gradually decreasing. In Figure 19b, the attention doing English reading comprehension is significantly greater in the 6000 k–500 lx and 5000 k–500 lx conditions than in the other two light environments. Furthermore, in Figure 19c, the attention to browsing articles is greatest in the light environment of 6000 k–500 lx. In Figure 19d, as the light environment weakens, the attention required to listen to music also weakens.

#### 4.3.1. Comparison of Different Light Environment for Four Tasks

The attention of the four tasks under different color temperature-illumination combinations is statistically analyzed using box plots. As shown in Figure 20, the box plot comparison results are generally consistent with the results presented in the raw data curves above.

Figure 20a depicts the attention of reciting English words. After the original data outlier cleaning, the concentration range is 100–45 under 6000 k–600 lx, 95–40 under 6000 k–500 lx, 85–20 under 5000 k–500 lx, and 70–10 under 3000 k–300 lx. The mean values of attention during word recitation at the four color temperature-illuminance combinations are shown in the figure: 6000 k–600 lx 71, 6000 k–500 lx 62, 5000 k–500 lx 51, and 3000 k–300 lx 45. Attention diminishes when the color temperature-illuminance decreases. The figure shows that the box widths of 6000 k–600 lx and 6000 k–500 lx are similar, and the data are relatively concentrated. The box width of 5000 k–500 lx is the widest, indicating that the data are the most spread out. The data of 3000 k–300 lx are concentrated but with the lowest mean attention value.

Figure 20b shows the attention level for doing English reading comprehension. Here, 50, 59, 56 and 42 are the average attention values under four color temperature-illumination combinations. The attention fluctuates widely at 6000 k–600 lx, and there are more outliers. It can be determined that the high color temperature-high illuminance combination is not suitable for doing English reading comprehension (RC). On the contrary, the high color temperature-medium illumination and medium color temperature-medium illumination combinations can maintain higher attention values, and the highest mean attention value is found in the 6000 k–500 lx light combination; the 3000 k–300 lx light combination makes the attention performance poor and has more outliers.

Figure 20c shows the attention of browsing essays. The mean values of attention under the four conditions of color temperature-illumination combinations are: 6000 k–600 lx 51, 6000 k–500 lx 65, 5000 k–500 lx 47, and 3000 k–300 lx 47. Similar to the attention for doing English reading comprehension, the 6000 k–500 lx combination is more suitable for browsing. Although the average attention value is maintained at 51 for the combination of high color temperature and high illumination, the large box width data are unfocused with large fluctuations and outliers. The average attention values for the medium color temperature-medium illumination and low color temperature-low illumination combinations are below 50, and there are many outliers in the 3000 k–300 lx illumination combination.

Figure 20d shows the attention values of listening to music. The average attention values of the four color temperature-illuminance combinations are 6000 k–600 lx 60, 6000 k–500 lx 53, 5000 k–500 lx 48, and 3000 k–300 lx 43. The average attention reduces with decreasing color temperature and illuminance at various combinations.

#### 4.3.2. Comparison of Different Tasks for Four Color Temperature-Illuminance Combinations

For further elaboration of the light environment’s effect on participants’ attention in different tasks, we compare the attention of the same color temperature-illumination combination. As shown in Figure 21, the mean value of attention for reciting English words reaches above 70 at 6000 k–600 lx, significantly higher than the other three tasks. This indicates that high color temperature-high illumination is more suitable for memory-based work scenarios. At 6000 k–500 lx, the mean attention values of reciting English words, performing English reading comprehension and browsing essays are above 60. This indicates that the combination of high color temperature-medium illumination has a facilitative effect on memory, thinking and understanding of work scenes. Under the combination of 5000 k–500 lx, the box plot median line is evenly distributed up and down. The attention levels of all four work tasks are at the medium level and the lowest attention level for listening to music. Under the combination of 3000 k–300 lx, the attention levels of reciting English words, doing English reading comprehension, browsing essays and listening to music are all below 50, indicating that the low color temperature-low illumination has a weakening effect on the concentration levels of memory, thinking, understanding, and leisure work situations, which makes the attention level unfocused.

Listening to music corresponds to a leisure scene at work, and listening to music in different light environments has a significant impact on attention and meditation. As shown in Figure 22, there is a significant contrast between attention and meditation when listening to music under different color temperature-illumination combinations. The mean value of attention tends to reduce with the decrease of color temperature-illumination, but the meditation tends to increase. Therefore, listening to music at the 3000 k–300 lx combination is least attentive and most relaxed.

## 5. Discussion

People work in various work scenarios, and providing the appropriate light environment for different work scenarios is crucial to enhancing work efficiency. In this paper, the IMU-based body posture capture system is used to reconstruct human posture. Compared to video and pressure sensor approaches, this method is easy to wear and does not interfere with the participant’s postural movements. It is not limited by space, light or range of motion during the acquisition process and allows kinematic analysis from the perspective of the whole body posture. In our self-developed wearable sensor-based posture capture system, the gradient descent method is used to fuse the sensor data and solve the posture. This is similar to the results of a typical optical motion capture system. The pelvis is a reference point to calculate the body’s posture iteratively. In practical experiments, the feature information of joint angles is used to recognize work scenarios, and the satisfactory classification results prove the practicality and stability of our posture capture method. In the future, we hope to upgrade sensor fusion techniques, which can significantly reduce the computational workload and improve the accuracy of posture updates.

Meanwhile, when the posture information is combined with EEG information for analysis, it reveals the changes in participants’ posture and attention under different lighting environments. In this paper, 4 × 4 experimental factors (4 postures × 4 color temperature-light combinations) are designed to discuss attention. The association between human posture and optimal light conditions in different work scenarios is established. Based on the study’s results, the optimal color temperature-light combinations for memory, thinking, understanding and leisure work scenes are determined. The memory work scenes are suitable to be performed at 6000 K–600 lx. The experimental results show that the thinking and understanding work scenarios have similar concentration requirements. Higher attention can be maintained in the combination of high color temperature-medium illumination and medium color temperature-medium illumination. Attention decreases in color temperature-illuminance combinations of 3000 K–300 lx. We should emphasize meditation rather than attention. The best results for meditation degrees are obtained at 3000 K–300 lx combinations, which help participants to reach the desired comfort state. Therefore, we believe that factors in different work scenarios are crucial to improving work efficiency. We also intend to apply our technology to intelligent offices and other areas to provide people with a pleasant working environment.

## 6. Conclusions

This paper presents a method to improve working based on wearable devices. A self-made posture detection system based on IMUs performs long-term human posture monitoring in a natural work environment. By assessing people’s attention and meditation levels under different work tasks and light conditions, we establish the association between different work states and optimal light conditions to enhance work efficiency. The effectiveness and accuracy of proposed attitude estimation algorithm have been verified through an optical motion capture system. Participants’ attention and meditation in different work tasks and light environments are monitored by the Mindwave EEG device. The experimental results show that this scheme can effectively identify human posture in various work tasks. Additionally, according to the feedback of attention level, we obtain the optimal light environment under different work tasks.

## Figures and Tables

**Figure 1 micromachines-13-01410-f001:**
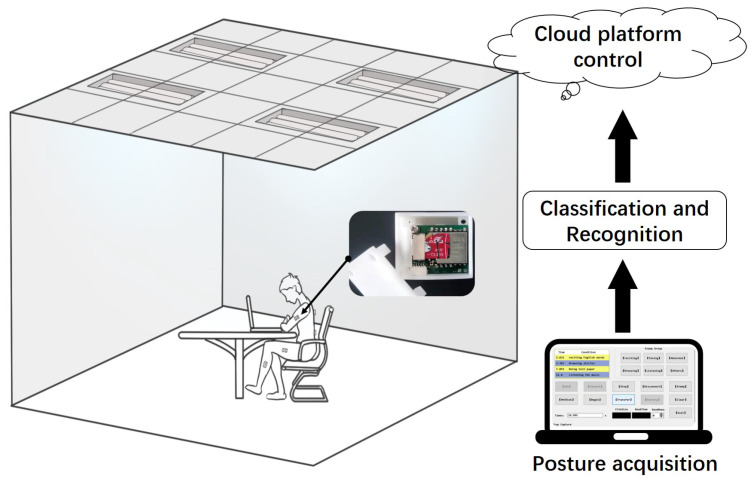
System platform architecture.

**Figure 2 micromachines-13-01410-f002:**
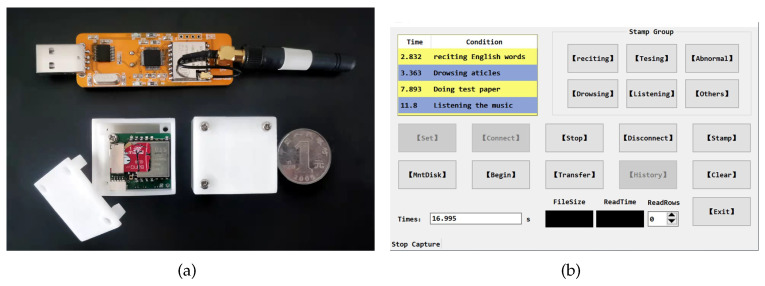
The hardware of the system platform and UI of control platform. (**a**) The hardware of the system platform consists of several self-made micro inertial sensor nodes and a transceiver. (**b**) The UI of the node control program is designed to manipulate the nodes to collect and convert posture data.

**Figure 3 micromachines-13-01410-f003:**
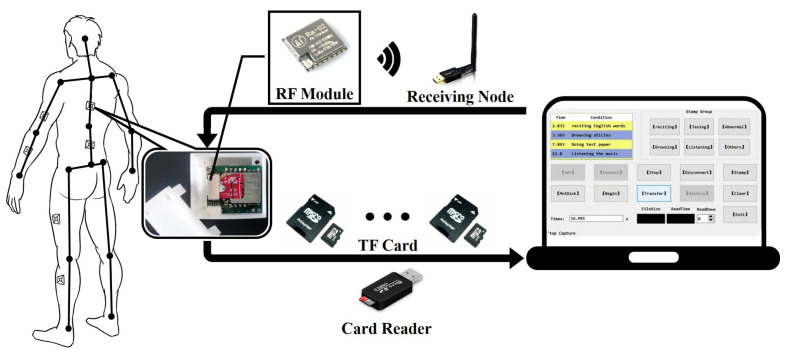
The framework of the posture capture system.

**Figure 4 micromachines-13-01410-f004:**
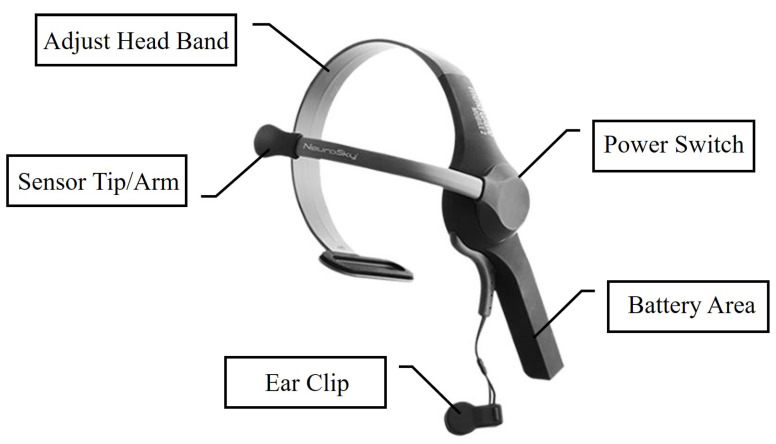
The shape of MindWave EEG device. It is worn on the head; the sensor tip is placed on the forehead, and the ear clip is clamped on the earlobe.

**Figure 5 micromachines-13-01410-f005:**
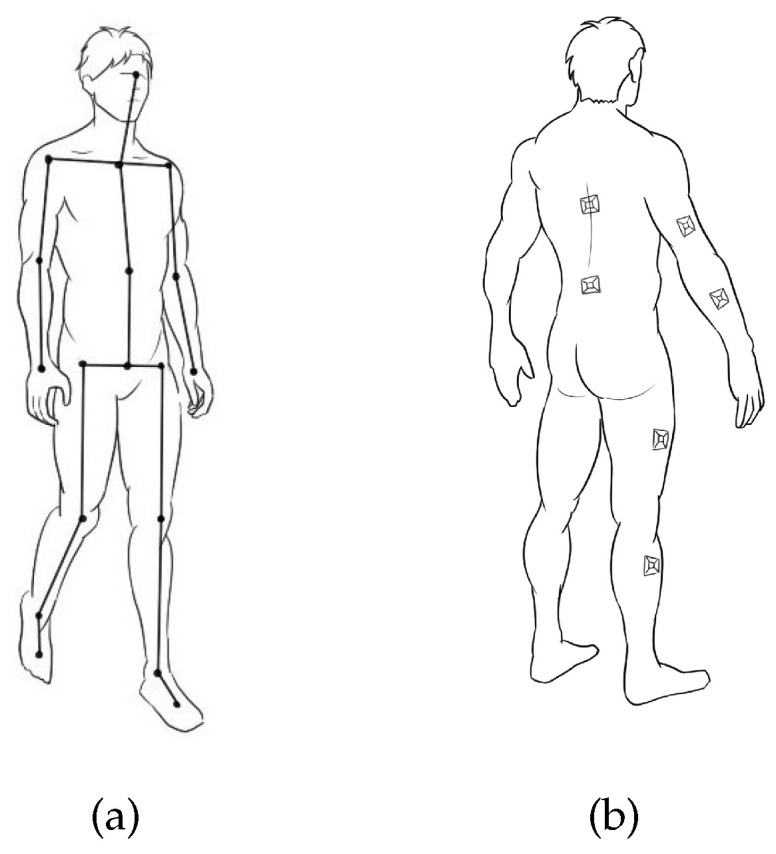
The mannequin definition. (**a**) The human body rigid model. (**b**) The position of inertial sensors on the body.

**Figure 6 micromachines-13-01410-f006:**
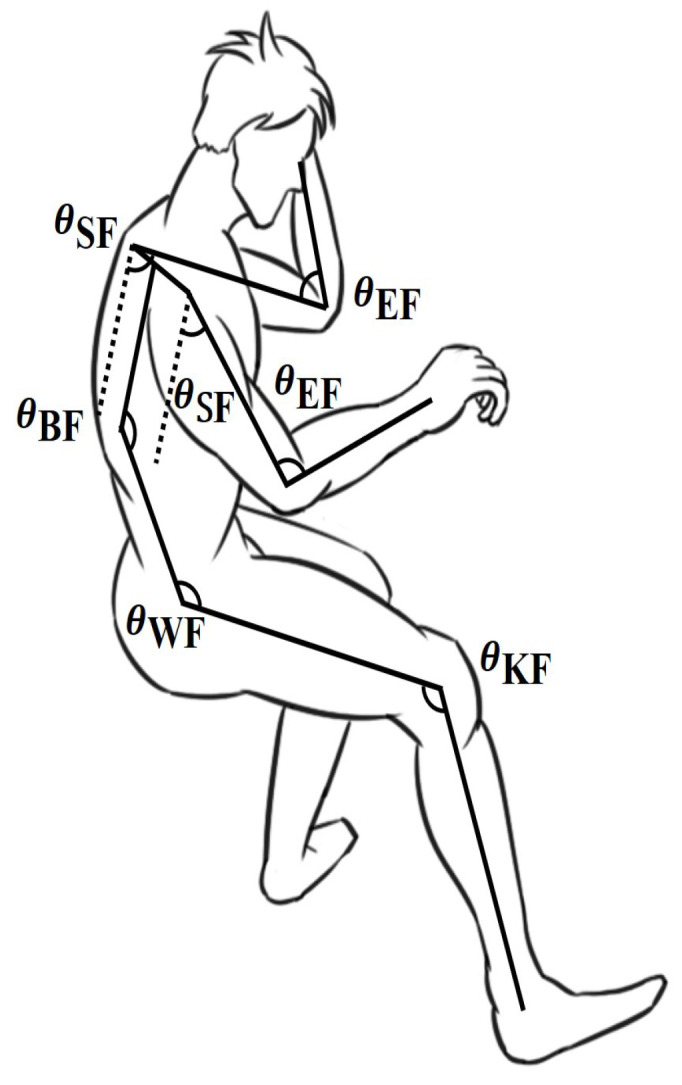
Definition of human joint angle.

**Figure 7 micromachines-13-01410-f007:**
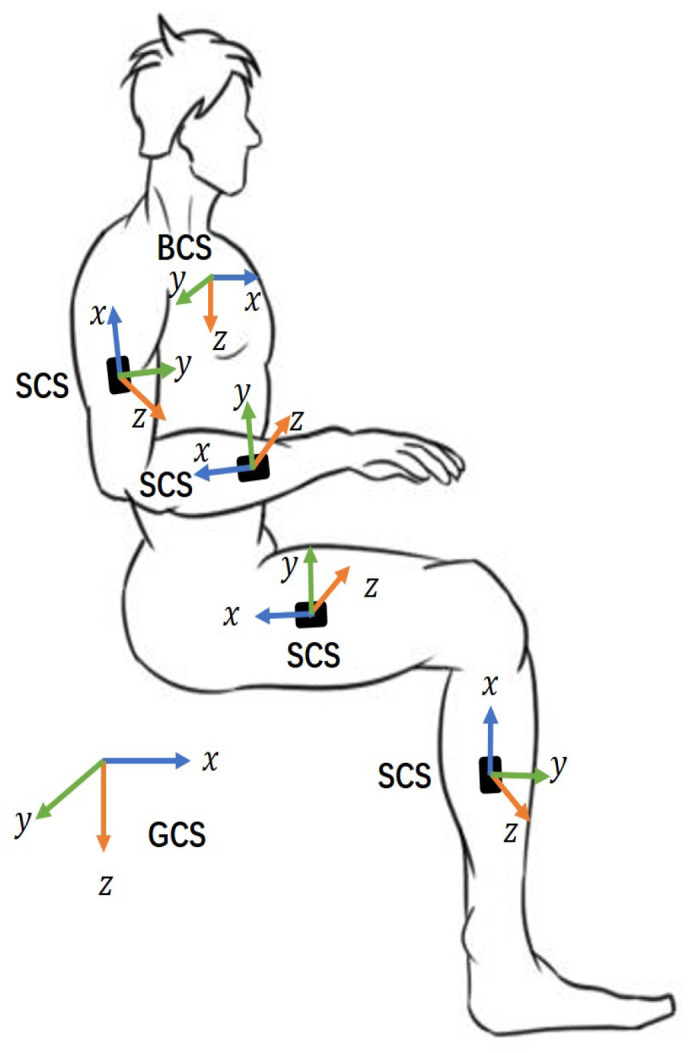
Definition of GCS, SCS, BCS.

**Figure 8 micromachines-13-01410-f008:**
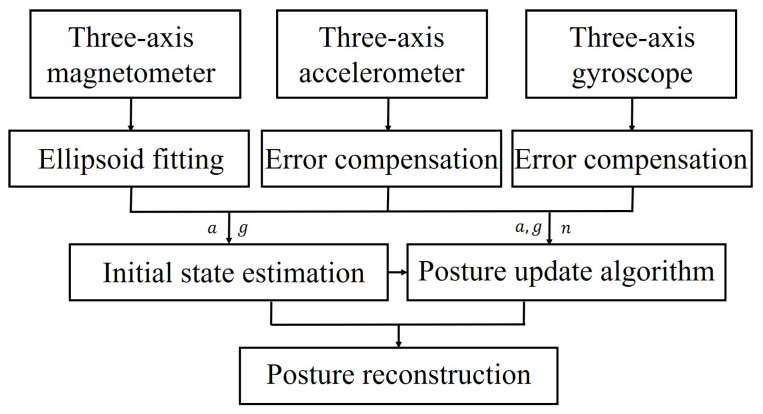
Motion analysis algorithm framework.

**Figure 9 micromachines-13-01410-f009:**
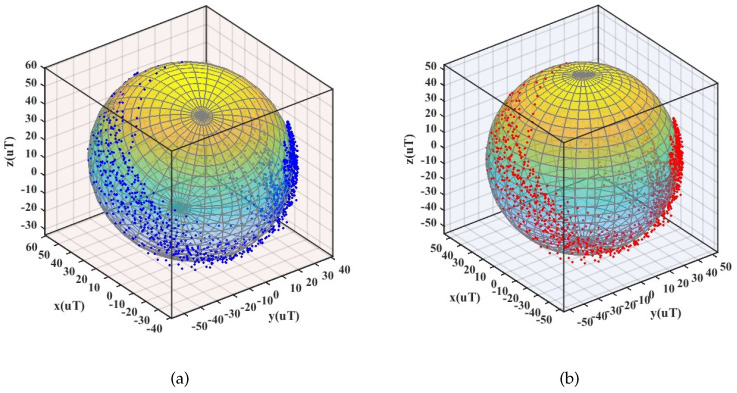
Magnetometer calibration: (**a**) before calibration (**b**) after calibration.

**Figure 10 micromachines-13-01410-f010:**
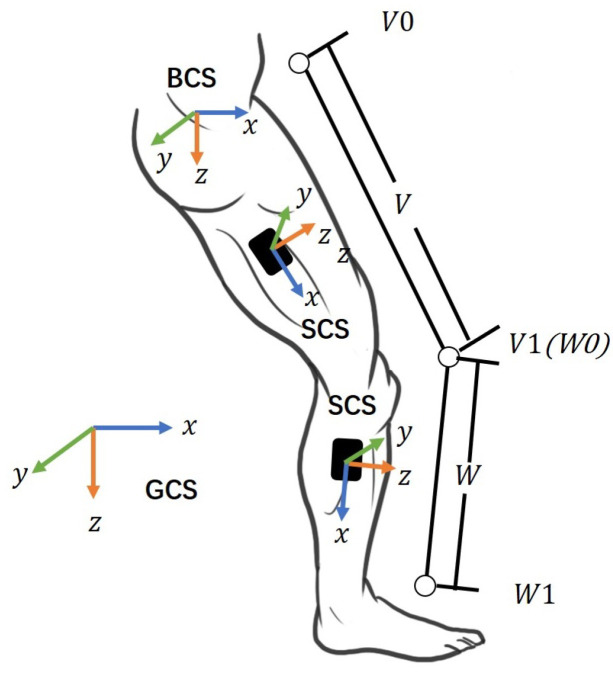
Leg limbs. The joint angle connects the thigh V and calf W.

**Figure 11 micromachines-13-01410-f011:**
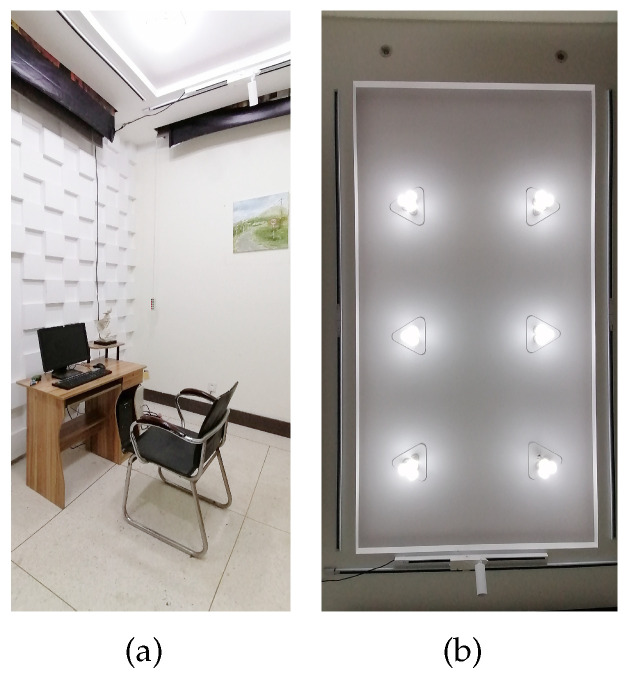
The experimental environment and lighting configuration. (**a**) The office environment of the Optical Experiment Base of Dalian Polytechnic University. (**b**) Light environment configuration with eighteen 9 W LED bulbs with adjustable illumination and color temperature.

**Figure 12 micromachines-13-01410-f012:**
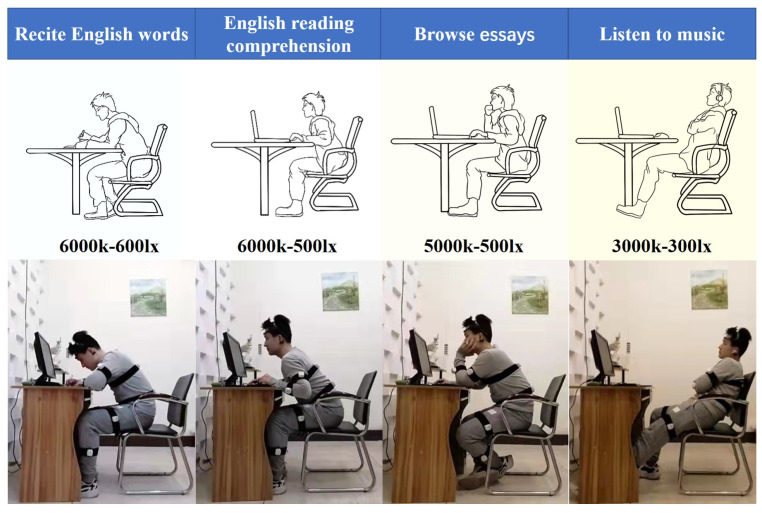
Four experimental tasks including four working postures (recite English words, English reading comprehension, browse essays and listen to music) and four light combinations (6000 k–600 lx, 6000 k–500 lx, 5000 k–500 lx, and 3000 k–300 lx).

**Figure 13 micromachines-13-01410-f013:**
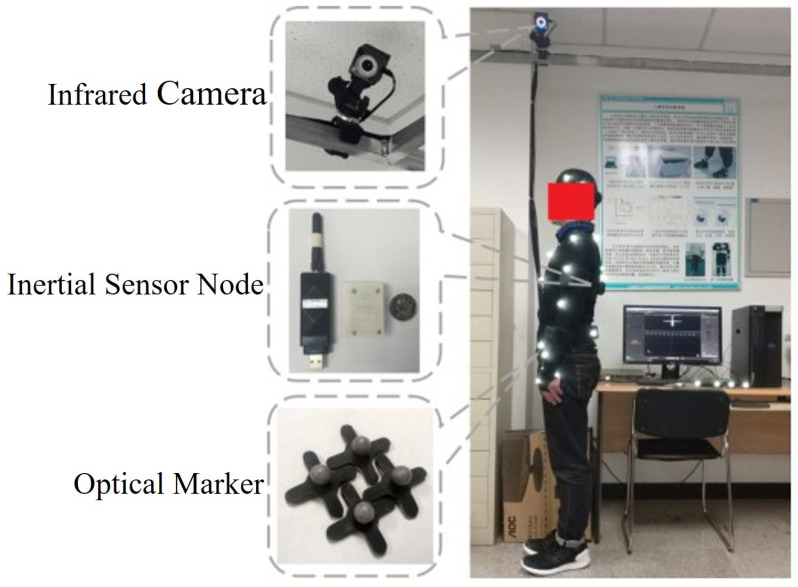
Contrast experimental scene between self-made system and optical capture system.

**Figure 14 micromachines-13-01410-f014:**
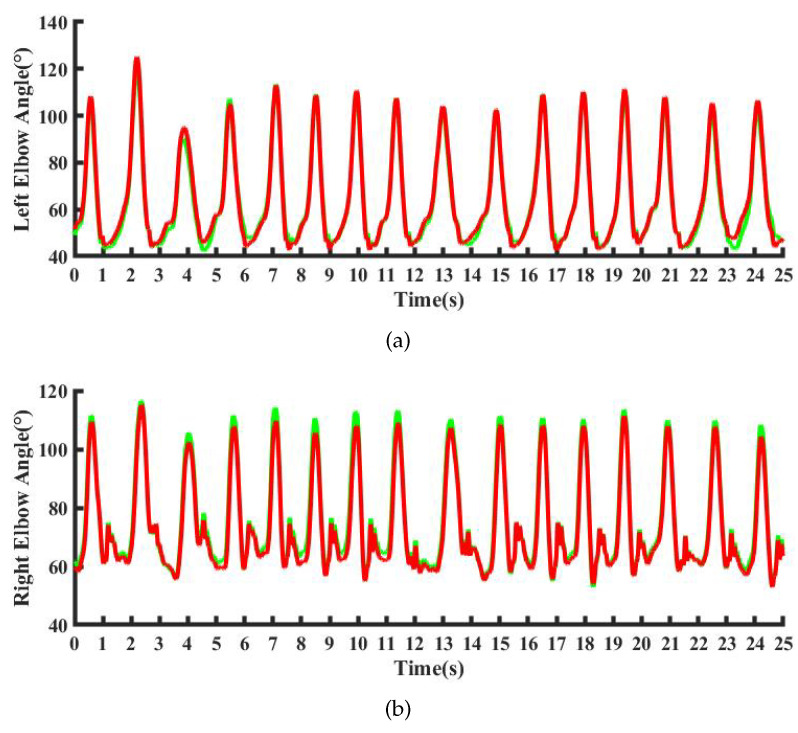
Comparison flexsion angle curve. (**a**) right elbow joint. (**b**) left elbow joints. The red curve is obtained by OptiTrack, and the green curve is obtained by the self-made system.

**Figure 15 micromachines-13-01410-f015:**
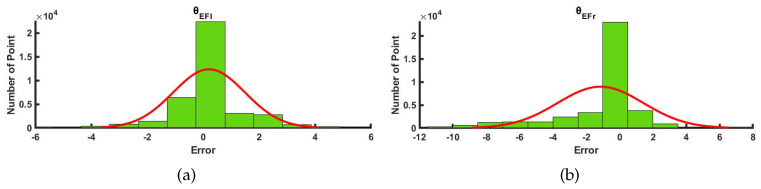
Error statistics of elbow joint angle. (**a**) Frequency histogram of left elbow joints. (**b**) Frequency histogram of right elbow joints.

**Figure 16 micromachines-13-01410-f016:**
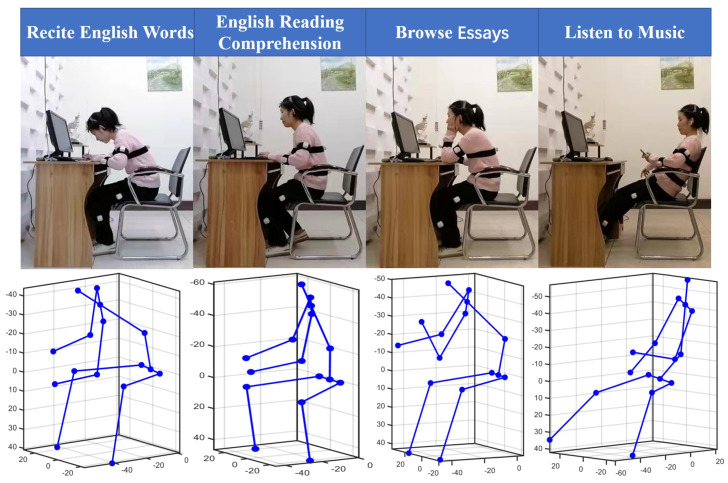
Participant’s posture of four tasks in the experiment.

**Figure 17 micromachines-13-01410-f017:**
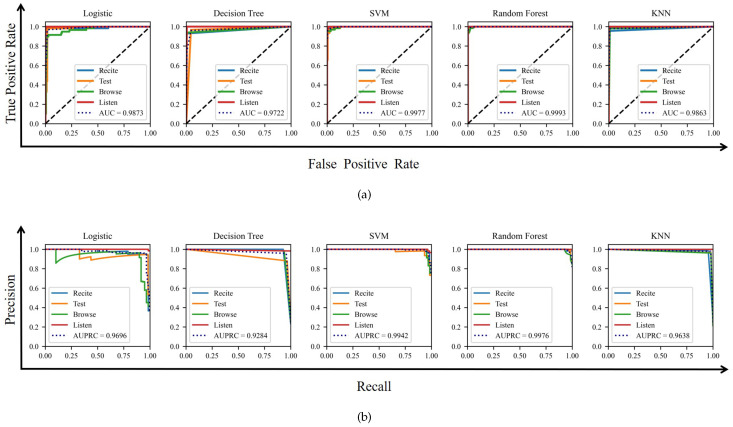
ROC and PRC of different classification models. (**a**) is ROC and (**b**) is PRC.

**Figure 18 micromachines-13-01410-f018:**
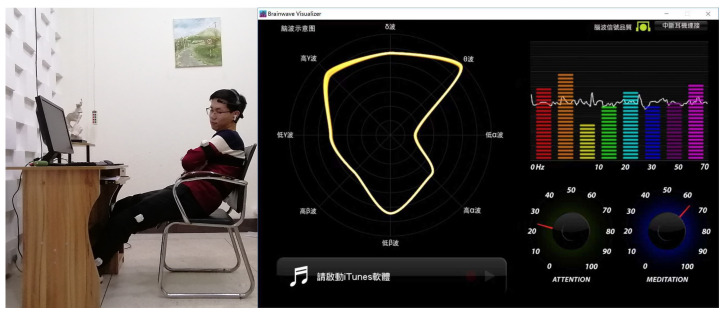
Brainwave Visualizer. The EEG signals, “Attention”, and “Meditation” of listening to music at 3000 k–300 lx.

**Figure 19 micromachines-13-01410-f019:**
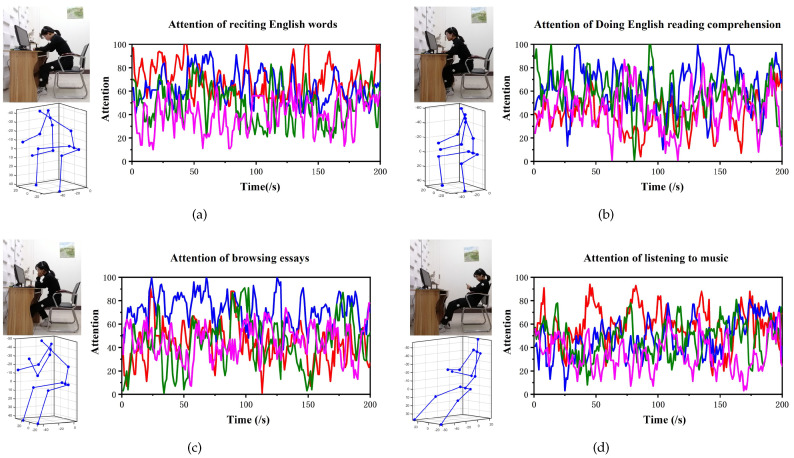
Raw data of concentration of 4 × 4 experimental conditions. (**a**) Attention of reciting English words. (**b**) Attention of doing English reading comprehension. (**c**) Attention of browsing essays. (**d**) Attention of listening to music. The **left** is the experimental task, and the **right** is the raw data of the participants’ brain waves.

**Figure 20 micromachines-13-01410-f020:**
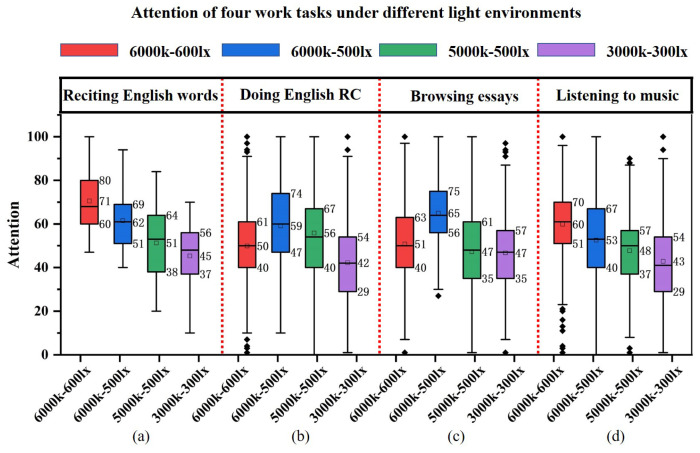
Box plot: attention of four work tasks in different light environments, including (**a**) reciting words, (**b**) doing reading comprehension, (**c**) browsing essays, and (**d**) listening to music.

**Figure 21 micromachines-13-01410-f021:**
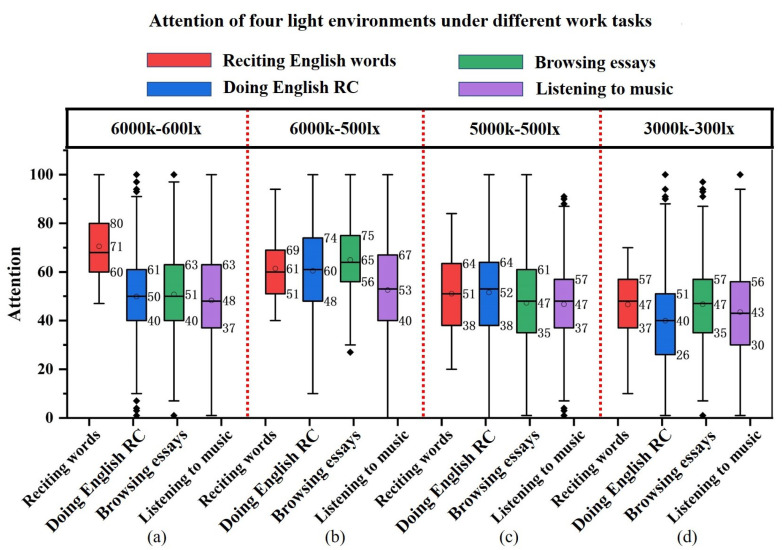
Box plot: attention of four light environments under different work tasks, including (**a**) 6000 k–600 lx, (**b**) 6000 k–500 lx, (**c**) 5000 k–500 lx and (**d**) 3000 k–300 lx.

**Figure 22 micromachines-13-01410-f022:**
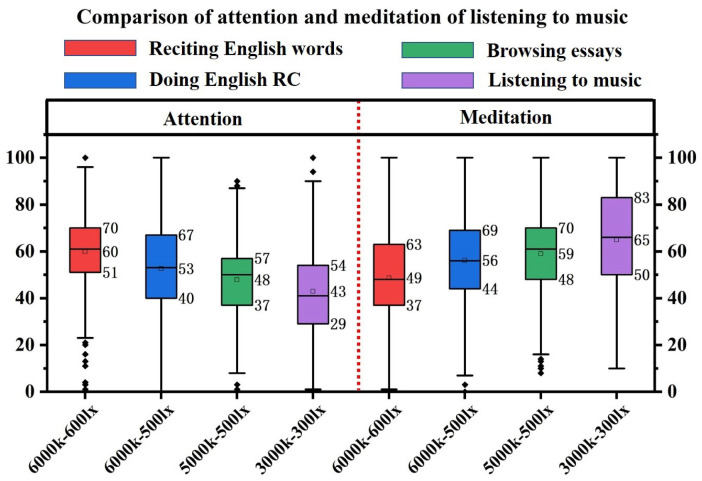
Box plot: Comparison of attention and meditation of listening to music.

**Table 1 micromachines-13-01410-t001:** IMUs internal structure parameters.

Unit	Accelerometer	Gyroscope	Magnetometer
Dimension	3 axis	3 axis	3 axis
Sensitivity (/LSB)	0.833 mg	0.04 deg/s	142.9 gauss
Dynamic Range	±18 g	±1200 deg/s	±1.9 gauss
Bandwidth (HZ)	330	330	25
Non-linear (%FS)	0.2	±0.1	0.1
Maladjustment (DEG)	0.2	0.05	0.25

**Table 2 micromachines-13-01410-t002:** MindWave performances specifications.

Performance	Parameter
Acquisition frequency band	0–100 Hz
Signal sampling frequency	512 Hz
ADC precision	12 digits
Dynamic response time	<2 ms
Environmental filtering	50/60 Hz
Data output frequency	1 Hz

**Table 3 micromachines-13-01410-t003:** 4 × 4 experimental conditions.

6000 K–600 lx Reciting words	6000 K–500 lx Reciting words	5000 K–500 lx Reciting words	3000 K–300 lx Reciting words
6000 K–600 lx Doing test paper	6000 K–500 lx Doing test paper	5000 K–500 lx Doing test paper	3000 K–300 lx Doing Test paper
6000 K–600 lx Browsing essays	6000 K–500 lx Browsing essays	5000 K–600 lx Browsing essays	3000 K–300 lx Browsing essays
6000 K–600 lx Listening to music	6000 K–500 lx Listening to music	5000 K–600 lx Listening to music	3000 K–300 lx Listening to music

**Table 4 micromachines-13-01410-t004:** Working condition feature table.

Feature	Name	Describe
Time domain feature	mean	The mean value of data
	med	The median value of data
	std	The standard deviation of data
	mad	The median absolute value of data
	quantile1	The 25th percentile of signal
	quantile2	The 75th percentile of signal
	iqr	Interquartile range
	skewness	The skewness of time signal
	kurtosis	The kurtosis of time signal
	var	The variance of time signal
	entropy	The entropy value of signal
	sepctral entropy	The sepctral entropy value of signal
Frequency domain feature	maxfreq	The maximum frequency of frequency features
	maxval	The maximum value of frequency features
	maxratio	The maximum ratio of frequency features
	peak	The main peak of autoregression features
	height	The second peak height of autoregression features
	position	The second peak position of autoregression features
	spwf	The spectral power features in 5 adjacent frequency

**Table 5 micromachines-13-01410-t005:** Predictive performance of different classification models.

Evaluation Metrics	Logistic Regression	Decision Tree	SVM	Random Forest	KNN
Accuracy	0.9665	0.9623	0.9707	0.9874	0.991
Precision	0.9668	0.9629	0.9714	0.9880	0.9917
Recall	0.9665	0.9623	0.9707	0.9874	0.9916
F1-score	0.9666	0.9623	0.9708	0.9875	0.9916
AUC	0.9893	0.9690	**0.9974**	**0.9998**	0.9883
AUPRC	0.9712	0.9216	**0.9921**	**0.9994**	0.9710

## Data Availability

Not applicable.

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
