# Peer review of "Research on the Efficiency of Working Status Based on Wearable Devices in Different Light Environments"

_micromachines, 2022, doi:10.3390/mi13091410_

Round 1
Reviewer 1 Report
Authors developed a wearable device for posture monitoring, and used EEG signal from MindWave for brain activity. It is a very interesting work. I recommend authors to address some minor points before accepting the manuscript.
1. please add some of the findings in terms of which classifier was best, why 15 IMU and 12 IR camera were used in the abstract rather than just saying what was done.
2. Please specify the sensors' name so that anyone can use and regenerate the data following the protocol. why 9 DOF IMU were used as many of the works just use 3 axis accelerometer for similar objective? Have you check any redundancy in the data?
3. Please elaborate the Figure 3 details? Lora wireless communication protocol sends command wirelessly and sensor acquire the posture data. instead of storing in TF card why not transfer data wirelessly again for remote processing in matlab?
4. Can you compare the efficiency of different classifier (5 classifier are used?) for posture detection?
5. Did you compare the posture capturing with IMU data and IR camera image data?
Author Response
我们要感谢审稿人以及副编辑,他们给我们提供了有益的评论指导。我们根据审查中提出的问题仔细修改了手稿。请参阅附件。

Reviewer 2 Report
This paper presents a wearable posture recognition system in an office environment. Different lighting conditions are provided as per the detected posture of the subject. The authors claim that proper lighting improves the performance of the workers. The idea of the paper is interesting, however, I have the following comments/suggestions to further improve the quality of the paper.
1. There are some minor grammatical errors and typos. The paper needs to be revised in terms of language. Also, the in-text citation style is not correct.
2. The main objective of the paper needs some clarifications. Is it to propose the posture detection system or to prove that various lighting condition has different effects on the human brain?
3. The related work needs some improvement. Some relevant works need to be presented and compared with the proposed system.
4. The proposed system is wearable-based, thus, it should be compared with a similar system for the performance evaluation.
5. I have a major concern with the practical use of the proposed system. It requires the user to wear numerous sensors (around 15). Is it even possible? What would be the cost, battery requirements, comfort level of the users, and any harmful effects on the human body? These details are missing from the paper.
6. Does the proposed system detect the user's posture in real-time or not? This detail is missing from the paper?
7. I also have concerns about the experimental evaluation of the proposed system. The authors have not provided many details such as the number of users, the duration of each experiment and each task, etc. The authors have used cross-validation for training the models. It is not clear whether the model needs to be trained for each individual user before it can predict its posture or can be trained on specific users and can then predict the posture of an unknown subject. The latter option is suitable for these types of applications and can be achieved by using Leave OneSubjectOut (LOSO).
8. Can the author justify the use of these many sensors in their proposed solution? Can we detect the user's posture using less number of sensors? There are some related works that can recognize human posture with fewer sensors.
Author Response

(The authors gave the same response as above.)

Round 2
Reviewer 2 Report
The authors have tried to address my previous comments and have improved the paper, however, I have the following comments.
1. There are still grammatical errors and typos, even in the abstract. The paper needs to be revised in terms of language. When referring to another work for the first time, it should be cited in the first sentence.
2. Abstract needs to be improved.
3. I am not satisfied with the author's response to this comment. "The proposed system is wearable-based, thus, it should be compared with a similar system for the performance evaluation.".
4. It is still not clear whether the model needs to be trained for each individual user before it can predict its posture or can be trained on specific users and can then predict the posture of an unknown subject.
5. The authors have still not provided any justification for using these many sensors in their proposed solution.
Author Response
We would like to thank the Reviewers, as well as the Associated Editor, for giving us guidance with their helpful comments. We have revised the manuscript carefully according to the issues raised in the review. Please see the attachment.
